# Four Different Finger Positions and Their Effects on Hemodynamic Changes during Chest Compression in Asphyxiated Neonatal Piglets

**DOI:** 10.3390/children10020283

**Published:** 2023-02-01

**Authors:** Marlies Bruckner, Mattias Neset, Megan O’Reilly, Tze-Fun Lee, Po-Yin Cheung, Georg M. Schmölzer

**Affiliations:** 1Centre for the Studies of Asphyxia and Resuscitation, Neonatal Research Unit, Royal Alexandra Hospital, Edmonton, AB T5H 3V9, Canada; 2Department of Pediatrics, Faculty of Medicine and Dentistry, University of Alberta, Edmonton, AB T6G 2E3, Canada; 3Division of Neonatology, Department of Pediatrics and Adolescent Medicine, Medical University of Graz, 8036 Graz, Austria

**Keywords:** infant, newborn, neonatal resuscitation, chest compression, asphyxia, 2 thumbs technique, 2 finger technique

## Abstract

**Background:** The Neonatal Life Support Consensus on Science With Treatment Recommendations states that chest compressions (CC) be performed preferably with the 2-thumb encircling technique. The aim of this study was to compare the hemodynamic effects of four different finger positions during CC in a piglet model of neonatal asphyxia. **Methods:** Seven asphyxiated post-transitional piglets were randomized to CC with 2-thumb-, 2-finger-, knocking-fingers-, and over-the-head 2-thumb-techniques for one minute at each technique. CC superimposed with sustained inflations were performed manually. **Results:** Seven newborn piglets (age 0–4 days, weight 2.0–2.1 kg) were included in the study. The mean (SD) slope rise of carotid blood flow was significantly higher with the 2-thumb-technique and over-the-head 2-thumb-technique (118 (45) mL/min/s and 121 (46) mL/min/s, respectively) compared to the 2-finger-technique and knocking-finger-technique (75 (48) mL/min/s and 71 (67) mL/min/s, respectively) (*p* < 0.001). The mean (SD) dp/dt_min_ (as an expression of left ventricular function) was significantly lower with the 2-thumb-technique, with −1052 (369) mmHg/s, compared to −568 (229) mmHg/s and −578(180) mmHg/s (both *p* = 0.012) with the 2-finger-technique and knocking-finger-technique, respectively. **Conclusion:** The 2-thumb-technique and the over-the-head 2-thumb-technique resulted in improved slope rises of carotid blood flow and dp/dtmin during chest compression.

## 1. Introduction

The Neonatal Life Support Consensus on Science With Treatment Recommendations (CoSTR) describes the 2-thumb encircling-technique (Figure 1A), and/or the 2-fingers-technique (Figure 1B), during chest compression (CC) [1]. Neonatal Resuscitation guidelines recommend preferably using the 2-thumb encircling-technique [2,3]. These recommendations are mainly based on manikin studies reporting that the 2-thumb encircling-technique results in increased (i) CC depth [4,5,6,7,8,9], (ii) chest release force or chest recoil [4,5], (iii) CC duty cycle [4], (iv) simulated blood pressures [10], and (v) correct finger position [6,11,12] compared to the 2-fingers-technique. In the 7th Edition of the Neonatal Resuscitation Program, the over-the-head 2-thumb-technique (Figure 1C), where the provider is positioned behind the newborn infant’s head, was introduced [13]. 

Several alternative CC hand/finger positions [14] have been described, including vertical thumbs [15], three-finger pinch [16], thumb and index finger [17], variations of the 2-finger-technique [18], flexed 2-fingers [19], 2-thumbs with fisted hands [20], or knocking-fingers-technique (Figure 1D) [21,22]. During the knocking-fingers-technique [21,22], the provider bends the proximal and distal interphalangeal joints by 90° and places the tip of the thumb against the palmar side of the middle phalanx of the index finger (Figure 1D).

However, except for one study [23], all of the remaining currently published studies were performed on manikins [14], which focused on CC performance and fatigue. We aimed to compare hemodynamic changes using the 2-thumb-, 2-finger-, knocking-fingers-, and over-the-head 2-thumb-techniques during CC in a post-transitional piglet model of neonatal asphyxia.

## 2. Materials and Methods

Term newborn mixed breed post-transitional piglets were obtained on the day of experimentation from the University Swine Research Technology Centre, University of Alberta, Edmoton, Canada. All experiments were approved by the Animal Care and Use Committee (Health Sciences), University of Alberta, AUP00002651, registered at preclincialtrials.eu (PCTE0000249), and conducted according to ARRIVE (Animal Research: Reporting of In Vivo Experiments) guidelines [24,25].

### 2.1. Animal Preparation

The model has been previously described with some modifications [26]. Following the induction of anesthesia using isoflurane, the piglets were tracheotomized and mechanically ventilated (Acutronic, Hirzel, Switzerland) with a respiratory rate of 16–20/min and pressure of 20/5 cmH_2_O. Oxygen saturation was kept within 90–100%. A 5-French Argyle^®^ (Klein-Baker Medical Inc. San Antonio, TX, USA) double-lumen catheter was inserted via the femoral vein for administration of fluids and medications. The glucose level and hydration were maintained with an intravenous infusion of 5% dextrose at 10 mL/kg/h. During the experiment, anesthesia was maintained with intravenous propofol: 5–10 mg/kg/h, and morphine: 0.1 mg/kg/h. After surgery, the piglets were stabilized for one hour [9].

### 2.2. Hemodynamic Parameters

The right common carotid artery was exposed and encircled with a real-time ultrasonic flow probe (2 mm; Transonic Systems Inc., Ithica, NY, USA) to measure the carotid blood flow as a surrogate for cardiac output. A 5-French Argyle^®^ single-lumen catheter was inserted above the right renal artery via the femoral artery for continuous arterial blood pressure monitoring using a Hewlett Packard 78833B monitor (Hewlett Packard Co., Palo Alto, CA, USA). A Millar catheter (MPVS Ultra, ADInstruments, Houston, TX, USA) was inserted into the left ventricle (LV) via the left common carotid artery for continuous measurement of stroke volume, end-diastolic volume, and left ventricular contractile function (dp/dt_max_, dp/dt_min_). Due to the size difference between the Millar catheter and LV longitudinal axis, which poses a limitation for the accuracy of in vivo volume measurement, an alpha factor = 0.46, based on comparison between Millar’s recording and direct echocardiographic measurements in three piglets, was used to correct the conductance volume [8,10]. 

### 2.3. Force Measurement

FlexiForce A201 sensors (TekScan, Boston, MA, USA) were placed on the piglets’ chests and on the fingers/thumbs that were used to perform CC. CC depth was measured with an infrared transmitter and receiver by placing the transmitter on the piglets’ chests and the receiver stationary on the resuscitation table. The applied CC force and depth was recorded with a sample rate of 200 Hz [27,28].

### 2.4. Experimental Protocol

Following surgical instrumentation and stabilization, piglets were exposed to 45 min of normocapnic hypoxia, followed by asphyxia, which was achieved by disconnecting the ventilator and clamping the endotracheal tube until asystole. Asystole was defined as zero carotid blood flow. Fifteen seconds after asystole, positive pressure ventilation was provided for 30 s with a Neopuff T-Piece (Fisher & Paykel, Auckland, New Zealand) with 21% oxygen, a peak inspiratory pressure of 30 cmH_2_O, a positive end-expiratory pressure of 5 cmH_2_O, and gas flow of 10 L/min. After 30 s of positive pressure ventilation, CC were started, with an anterior-posterior chest diameter depth of one third [27,28], using continuous CC during sustained inflation with a peak inspiratory pressure of 30 cmH_2_O. Each sustained inflation was 30 s long and interrupted for 1 s before a further 30 s long sustained inflation was provided. The rate of CC was 90/min. This technique was performed during whole CPR. This CC technique provides continuous CC, which are superimposed by high distending pressure (=sustained inflation) and allows for passive ventilation with each CC [29,30]. Throughout the study sequence 21% oxygen was used [31]. The sequence of 2-thumbs-, 2-fingers-, knocking-fingers-, and over-the-head 2-thumbs-techniques, was randomized in all piglets using a computer-generated randomization program. Sequentially numbered, sealed, brown envelopes containing the order of CC techniques were opened during the experiment. CC were performed manually by the same provider (GMS) for a duration of one minute for each technique and was blinded to real-time feedback [12,13].

### 2.5. Data Collection and Analysis

The demographics of the study piglets were recorded. Transonic flow probes and pressure transducer outputs were digitized and recorded with the LabChart^®^ programming software (ADInstruments, Houston, TX, USA). Ten second hemodynamic recordings immediately before hypoxia (baseline) and during each technique (20 s after starting compression) were used for comparison. The slope of the rise in carotid flow was used as a surrogate for cardiac output. Arduino software (Arduino, Somerville, MA, USA) was used to record depth data. Continuous variables are presented as means (standard deviation (SD)). The data were tested for normality (Shapiro-Wilk and Kolmogorov-Smirnov test) and compared using one-way repeated measures ANOVA with Tukey test for post hoc analysis. Statistical analyses were performed with SigmaPlot (Systat Software Inc., San Jose, CA, USA). 

## 3. Results

A total of seven post-transitional piglets were included, with a median (range) age of 3 (0–4) days old and weight of 2.0 (1.8–2.1) kg. Three piglets (43%) were female. Data for CC force, CC depth, hemodynamic, and respiratory parameters during CC are presented in Table 1. There was no return of spontaneous circulation during CC.

The mean (SD) slope rise of the carotid blood flow was significantly higher with the 2-thumb-technique and the over-the-head 2-thumb-technique (118 (45) mL/min/s and 121 (46) mL/min/s, respectively), compared to the 2-finger-technique and knocking-finger-technique (75 (48) mL/min/s and 71 (67) mL/min/s, respectively) (*p* < 0.001) (Table 1). 

Stroke volume, as a percentage from baseline, was significantly improved with the 2-thumb-technique compared to the 2-finger-technique (Table 1), while it was not significantly different to the knocking-finger-technique or the over-the-head 2-thumb-technique (Figure 1E). 

Similarly, the mean (SD) dp/dt_min_ was significantly lower with the 2-thumb-technique, with −1052 (369) mmHg/s, compared to −568 (229) mmHg/s and −578 (180) mmHg/s (both *p* = 0.012) with the 2-finger-technique and knocking-finger-technique, respectively (Table 1, Figure 1H). dp/dt_min_ with the over-the-head 2-thumb-technique was −711 (310) mmHg/s, which was not different from the 2-thumb-technique (Table 1, Figure 1H).

## 4. Discussion

In this study, we compared four different finger/hand positions during neonatal CC, as the optimal finger/hand position during neonatal resuscitation remains unknown.

A randomized manikin study reported the correct CC depth in 99–100% (2-thumb-technique), 93–100% (knocking-fingers-technique), and 53–98% (2-fingers-technique) (*p* < 0.001) [21]. Similarly, Rodriguez-Ruiz et al. randomized healthcare professionals to perform CC on an infant manikin with either a new-two-thumb-technique, knocking-fingers-technique, or the two-thumb-techniques [22]. While not statistically significant, the correct median (IQR) CC depth was more likely achieved with the new-two-thumb-technique [96 (89.7–100)] or the two-thumb-techniques [98 (77.5–100)], compared to the knocking-fingers-technique [89.5 (57–99)] [22]. In an animal model, Houri et al. reported a significantly higher sternal compression force with the 2-thumb-technique compared to the 2-fingers-technique in pediatric asphyxiated piglets (22.9 vs. 14.6 psi, *p* < 0.05) [23]. Our downward force was similar with all techniques (Table 1), however, the CC depth was higher with both 2-thumb-techniques, with similar CC depth with the knocking-fingers-technique and the 2-finger-technique. This suggests that providers might achieve more CC depth with either of the 2-thumb-techniques. Our results are supportive of previous simulations and animal studies suggesting that any of the 2-thumb-techniques provide an improved CC depth compared to using the 2-finger-technique. While clinical data are lacking, and it may be rather unrealistic to expect a clinical trial comparing the 2-thumb-techniques with the 2-finger-technique, the available knowledge from manikin and animal data might be sufficient to support the current recommendations [1,2].

Dorfsmann et al. compared the 2-thumb- and 2-finger-techniques using a modified manikin, with a 50 mL bag of normal saline solution below the chest plate attached to an arterial pressure transducer [10]. Overall, the 2-thumb-technique had improved arterial blood pressure [systolic (68.9 vs. 44.8 mmHg), diastolic (17.6 vs. 12.5 mmHg), and mean (35.3 vs. 23.3 mmHg)]. However, a randomized trial in asphyxiated pediatric piglets comparing the 2-thumb-technique with the 2-finger-technique reported significantly higher systolic (22.7 vs. 14.5 mmHg, *p* = <0.05), but similar diastolic (3.5 vs. 3.4 mmHg) blood pressure with the 2-thumb-technique [23]. Further, Menegazzi et al. reported systolic, diastolic, mean arterial, and coronary perfusion pressures were higher with the 2-thumb-technique compared to the 2-finger-technique [32]. In the current study, the 2-thumb-technique and over-the-head 2-thumb-technique generated significantly faster slope rise of carotid blood flow (*p* = 0.0001), and significantly lower dp/dt_min_ (*p* = 0.003) compared to the 2-fingers- and knocking-fingers-techniques (Table 1), suggesting improved left ventricular function with both 2-thumb-techniques. The absence of an increase in carotid blood flow, despite a faster rise of carotid blood flow during CC, might be explained by the applied CC rate. A higher CC rate results in a decreased time interval between the compressions, which might have kept the starting advantage of the left ventricular ejection when using either of the two 2-thumb-techniques. This could have resulted in a more constant and increased flow. Hemodynamic changes might have also been affected by differences in duty cycles between the CC techniques, since a lower duty cycle (faster compression and slower decompression) leads to increased hemodynamic parameters (e.g., coronary perfusion pressure and arterial blood pressure). However, we did not measure the duty cycle during the experiment, hence, we cannot rule out an impact of different duty cycles when providing any of the techniques.

There is a lack of data comparing the 2-thumb-technique and the over-the-head 2-thumb-technique. Cheung et al. reported that the quality of CC was not different when participants performed CPR with either the 2-thumb-technique or the over-the-head 2-thumb-technique [33]. Furthermore, most participants (87%) liked the CC performed using the over-the-head 2-thumb-technique [33]. In comparison, Jo et al. compared the over-the-head 2-thumb-technique with the 2-fingers-technique and reported a greater CC depth, more effective CC, complete recoil, and a lower fatigue score with the over-the-head 2-thumb-technique [34]. Our results confirm these observations from manikins, as we observed no difference between the 2-thumb-technique and the over-the-head 2-thumb-technique, while the 2-fingers-technique had a lower slope rise of carotid blood flow, stroke volume, and dp/dt_min_.

While the LV dp/dt min was not different between both 2-thumb-techniques, it was significantly lower with the 2-fingers- and knocking-fingers-techniques, which indicated worse ventricular relaxation. As the CC rate was not different between groups, we speculate that with the 2-fingers- and knocking-fingers-techniques the provider was not generating adequate compression force when compared to the 2-thumb-techniques. This would be in agreement with similar findings in changes of stroke volume and slope rise of carotid blood flow. 

In the current study we used an alternative approach to chest compression, which is mentioned in the knowledge gap of neonatal consensus of science and treatment recommendations [35]. During this technique, chest compressions are superimposed with a sustained inflation (SI) (CC + SI) [30], which results in passive lung aeration (improved minute ventilation and oxygenation) and significantly higher pulmonary and carotid blood flow [30]. In a post-translational piglet model, CC + SI compared to 3:1 C:V resulted in a significantly reduced time to return of spontaneous circulation with 38 (23–44) s vs. 143 (84–303) s (*p* = 0.0008)], and improved survival [7/8 [87.5%] vs. 3/8 [37.5%] (*p* = 0.038)] [29]. In a small randomized pilot trial in preterm infants, comparing CC + SI with 3:1 C:V resulted in a significantly improved return of spontaneous circulation (CC + SI group 31 (9) s vs. 138 (72) s with 3:1 C:V group (*p* = 0.011)) [36]. Most recently, we have completed the SURV1VE-trial, a multi-center, cluster randomized trial comparing CC + SI with 3:1 C:V during CC in the delivery room [30,37], which we aim to publish the results from in 2023.

The current neonatal resuscitation guidelines recommend 100% oxygen during CC [1,2], which is based on expert opinion and extrapolations from animal studies. Several animal studies comparing 21% and 100% oxygen during chest compression reported similar rates of mortality and time to return of spontaneous circulation [38,39]. Furthermore, in a meta-analysis of eight animal studies (*n* = 323) comparing 21% and 100% oxygen during CC, Garcia-Hidalgo et al. reported no difference in rates of mortality (risk ratio 1.04 [0.35, 3.08], I^2^ = 0%, *p* = 0.94) and/or time to return of spontaneous circulation (mean difference −3.8 [−29.7–22] s, I^2^ = 0%, *p* = 0.77) [31]. While these animal data support the use of 21% oxygen during CC, no human study has examined this. In the current study we used 21% oxygen, which is different to the current recommendations; while this is a limitation, the purpose of the study was to examine the hemodynamic effect of different finger/hand positions in a non-surviving animal model. 

### Limitations

Although CC + SI [30] is mentioned in the knowledge gap section of the neonatal resuscitation guidelines, it is currently not a recommended treatment option [1,2]. CC + SI might positively or negatively affect venous return, cardiac transmural, and thoracic pressure gradients [30]. Using the current recommended CC approach of 3:1 compression:ventilation ratio might yield different results [1,2]. We used a piglet asphyxia model that closely simulates delivery room events, with a gradual onset of severe asphyxia leading to asystole. However, our piglets had already undergone the fetal-to-neonatal transition, were sedated/anesthetized, and used tracheostomy with a tightly sealed endotracheal tube, which does not occur in the delivery room. The aim of the study was to examine the hemodynamic effects of four different finger positions, therefore these limitations are less likely to have influenced the results. The different shape and surface anatomy of the piglet chest to that of humans might have generated different forces, and therefore a different hemodynamic response. However, a piglet has a similar anatomy to humans and a similar chest size and shape, and performing CC on a piglet feels very similar to CPR in newborn infants [40,41], which is essential for the current study.

## 5. Conclusions

The 2-thumb-technique and the over-the-head 2-thumb-technique resulted in improved slope rise of carotid blood flow and dp/dtmin during chest compression. Future animal and clinical studies should examine the optimal finger/hand position to improve outcomes during neonatal cardiopulmonary resuscitation.

## Figures and Tables

**Figure 1 children-10-00283-f001:**
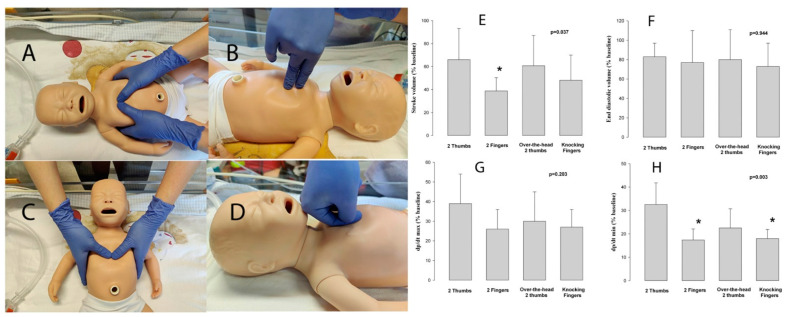
Chest compression techniques: (**A**) 2-thumb-technique, (**B**) 2-finger-technique, (**C**) over-the-head 2-thumb-technique, (**D**) knocking-fingers-technique, (**E**) percentage changes of stroke volume, (**F**) end diastolic volume, (**G**) maximal rate of rise of left ventricular pressure (dp/dt max), and (**H**) minimum rate of change of ventricular pressure (dp/dt min) to baseline. * Significantly different from 2-thumbs-technqiue (*p* < 0.05, Tukey).

**Table 1 children-10-00283-t001:** Force, Depth, Hemodynamic, and Respiratory parameters during chest compression.

	Baseline Parameters	2-Thumb-Technique (n = 7)	2-Finger-Technique (n = 7)	Over-the-Head 2-Thumb-Technique (n = 7)	Knocking-Fingers-Technique (n = 7)	*p*-Value
Applied Force (kg)		1.30 (0.54)	1.23 (0.62)	1.04 (0.51)	1.29 (0.66)	0.594
Applied CC Depth (cm)		3.3 (1.0)	2.3 (1.0)	3.3 (1.6)	2.4 (1.1)	0.325
Anterior-posterior CC depth (%)		38 (0)%	25 (0)%	38 (0)%	26 (0)%	
	**Hemodynamic Parameter**					
Carotid blood flow (mL/kg/min)	0 (0)	10 (6)	5 (3)	6 (5)	4 (3)	0.13
Slope rise of carotid blood flow (mL/min/s)	0 (0)	118 (45)	75 (48)	121 (46)	71 (67)	0.001
Mean arterial blood pressure (mmHg)	0 (0)	19 (9)	10 (5)	12 (5)	12 (7)	0.12
Diastolic blood pressure (mmHg)	0 (0)	9 (4)	8 (2)	8 (2)	8 (2)	0.67
Stroke volume (mL/kg)	0 (0)	0.8 (0.3)	0.5 (0.2) *	0.8 (0.3)	0.6 (0.3)	0.12
End diastolic volume (mL/kg)	0 (0)	2.6 (1.4)	2.3 (1.3)	2.4 (1.5)	2.1 (1.2)	0.94
dp/dt_max_ (mmHg/s)	0 (0)	1128 (405)	790 (398)	877 (478)	796 (357)	0.40
dp/dt_min_ (mmHg/s)	0 (0)	−1052 (369)	−568 (229) *	−711 (310)	−578 (180) *	0.012
	**Respiratory Parameter**					
Tidal volume (mL/kg)		9.5 (3.6)	8.5 (3.5)	7.6 (1.9)	7.8 (2.4)	0.611
Minute Ventilation (mL/kg/min)		855 (320)	763 (313)	680 (172)	702 (2019)	0.432
Peak Inspiratory Flow (L/min)		5.7 (1.6)	5.5 (0.8)	4.7 (0.3)	5.0 (0.9)	0.291
Peak Expiration Flow (L/min)		−9.1 (1.5)	−8.7 (3.0)	−7.4 (1.7)	−9.3 (1.8)	0.337
Peak Inflation Pressure (cm H_2_O)		29 (10)	30 (13)	28 (12)	30 (11)	0.996
End-tidal CO_2_ (mmHg)		2.6 (2.8)	1.6 (1.2)	1.9 (1.7)	1.6 (1.0)	0.676
Rate (/min)		90 (1)	90 (1)	90 (1)	90 (1)	1.000

Data are presented as mean (SD), maximal rate of rise of left ventricular pressure (dp/dt_max_), minimum rate of change of ventricular pressure (dp/dt_min_), Rate = Ventilation and number of chest compressions, which corresponds with the number of ventilations per min. * Significantly different from the 2-thumb-technique group (Tukey).

## Data Availability

All data is available wihtin the manuscript.

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
