# Peer review of "Four Different Finger Positions and Their Effects on Hemodynamic Changes during Chest Compression in Asphyxiated Neonatal Piglets"

_children, 2023, doi:10.3390/children10020283_

Round 1
Reviewer 1 Report
This is an excellent paper of high relevance for the clinical management of resuscitation.
There are no changes required. The paper can be accepted without changes.
Congratulations!
Author Response
Thank you
Reviewer 2 Report
Bruckner and colleagues present a study comparing 4 different methods of chest compressions in piglet cardiac arrest animal model. Seven piglets have been studied. Hemodynamic parameters, depth of compression, force applied have been compared. The study concludes that the 2-thumb technique and over the head 2-thumb technique resulted in improved hemodynamics during compression.
I have the following comments and questions.
Introduction:
1. The manuscript provides adequate introduction to the topic of the study.
Methods
1. What was the rate of compression during CPR
2. The main difference in the two thumb and over the head 2 thumb technique demonstrated in Figure 1 seems to be the additional compression of the liver in Figure 1 A, which could potentially increase venous return to the heart. Does the piglet model used replicate this feature of human anatomy well? If not they seem to be essentially same.
3. Was the EKG monitored during compressions? Was there any ROSC during the studies?
Results:
1. What was the mean blood pressure of the animals prior to the arrest? Would be a interesting comparison if the hemodynamics prior to the arrest were listed in Table 1.
2. Was the CVP monitored?
3. Statistical analysis needs review by a statistician.
4. Based on the results reported in Table 1. only carotid flow and end diastolic volume (ml/kg) dp/dt (mmHg/s) seem to be significantly different between the groups.
5. Would be best to avoid statements like "the stroke volume was significantly improved but was not statistically significant" (Page 4: line 144-146)
Conclusion:
1, The statistical analysis doesn't seem to support the stated conclusion.
Summary:
A well designed study that seeks to objectively compare different techniques of chest compression during cardiac arrest. The statistical analysis needs further review. The conclusion stated doesn't seem to be supported by the statistical analysis
Author Response
Bruckner and colleagues present a study comparing 4 different methods of chest compressions in piglet cardiac arrest animal model. Seven piglets have been studied. Hemodynamic parameters, depth of compression, force applied have been compared. The study concludes that the 2-thumb technique and over the head 2-thumb technique resulted in improved hemodynamics during compression.
I have the following comments and questions.
Introduction:
- The manuscript provides adequate introduction to the topic of the study.
Response: Thank you
Methods
- What was the rate of compression during CPR
Response: the rate of compression was 90 per minute. This has been added to the method section.
- The main difference in the two thumb and over the head 2 thumb technique demonstrated in Figure 1 seems to be the additional compression of the liver in Figure 1 A, which could potentially increase venous return to the heart. Does the piglet model used replicate this feature of human anatomy well? If not they seem to be essentially same.
- Was the EKG monitored during compressions? Was there any ROSC during the studies?
Response: Yes, the EKG, along with the invasive monitoring was monitored throughout the study by a dedicated researcher. There was no ROSC during the experiment, this has been added to the result section.
Results:
- What was the mean blood pressure of the animals prior to the arrest? Would be a interesting comparison if the hemodynamics prior to the arrest were listed in Table 1.
Response: They were zero for all parameters, as we waited to total cardiac arrest to prevent any interference with spontaneous heart activity. We have added them to Table 1.
- Was the CVP monitored?
Response: The CVP was measured and monitored, but not recorded for analysis.
- Statistical analysis needs review by a statistician.
- Based on the results reported in Table 1. only carotid flow and end diastolic volume (ml/kg) dp/dt (mmHg/s) seem to be significantly different between the groups.
Response: this has been edited.
- Would be best to avoid statements like "the stroke volume was significantly improved but was not statistically significant" (Page 4: line 144-146)
Response: this has been edited.
Conclusion:
1, The statistical analysis doesn't seem to support the stated conclusion.
Response: this has been edited.
Summary:
A well designed study that seeks to objectively compare different techniques of chest compression during cardiac arrest. The statistical analysis needs further review. The conclusion stated doesn't seem to be supported by the statistical analysis
Response: Thanks
Reviewer 3 Report
This study aims at comparing the hemodynamic effects of four different finger positions during chest compressions (CC) in a piglet model of neonatal asphyxia. Seven asphyxiated post-transitional piglets were randomized to CC with 2-thumb, 2-finger, knocking fingers and over-the-head-2-thumb techniques for 1 minute at each technique. Studied parameters were: 1) hemodynamic parameters: carotid blood flow, arterial blood pressure, stroke volume, end-diastolic volume, and left ventricular contractile function; 2) force measurements: applied CC force and depth. During CC, albeit force parameters were similar between techniques, the 2-thumb and the over-the-head-2-thumb techniques resulted in improved hemodynamics compared with the 2-finger and knocking finger techniques.
My comments are:
Page 2, Fig. 1 D – The knocking-finger position is not clear, it would be better to change it with a picure that can better illustrate the correct position of the thumb and of the index
Page 2, line 73 – The 5-French Argyle double-lumen catheter was inserted in the umbilical vein?
Discussion:
Limitations: although limitations are well addressed in this section, the current recommended approach suggests to use 100% oxygen while performing CC. Do the author think that using 100% oxygen during CC, instead of 21% as per study protocol, might influence results? Why the authors did not follow in their study the recommended approach as per 2020 International Guidelines (i.e. 3:1 compression:ventilation ratio and 100% oxygen)? Please comment on these topics in the discussion
Author Response
This study aims at comparing the hemodynamic effects of four different finger positions during chest compressions (CC) in a piglet model of neonatal asphyxia. Seven asphyxiated post-transitional piglets were randomized to CC with 2-thumb, 2-finger, knocking fingers and over-the-head-2-thumb techniques for 1 minute at each technique. Studied parameters were: 1) hemodynamic parameters: carotid blood flow, arterial blood pressure, stroke volume, end-diastolic volume, and left ventricular contractile function; 2) force measurements: applied CC force and depth. During CC, albeit force parameters were similar between techniques, the 2-thumb and the over-the-head-2-thumb techniques resulted in improved hemodynamics compared with the 2-finger and knocking finger techniques.
My comments are:
Page 2, Fig. 1 D – The knocking-finger position is not clear, it would be better to change it with a picure that can better illustrate the correct position of the thumb and of the index
Response: Thanks, we have updated the image
Page 2, line 73 – The 5-French Argyle double-lumen catheter was inserted in the umbilical vein?
Response: it was inserted in the femoral vein. This has been added
Discussion:
Limitations: although limitations are well addressed in this section, the current recommended approach suggests to use 100% oxygen while performing CC. Do the author think that using 100% oxygen during CC, instead of 21% as per study protocol, might influence results? Why the authors did not follow in their study the recommended approach as per 2020 International Guidelines (i.e. 3:1 compression:ventilation ratio and 100% oxygen)? Please comment on these topics in the discussion
Response: Thank you, we have added this to the discussion.
Round 2
Reviewer 2 Report
Thank you for submitting the revised manuscript. I have no further comments.